# TANGO: Text-driven Photorealistic and Robust 3D Stylization via Lighting Decomposition

**Yongwei Chen[1,3], Rui Chen[1], Jiabao Lei[1], Yabin Zhang[2], Kui Jia[1,4]***

[1]South China University of Technology [2]The Hong Kong Polytechnic University
[3]DexForce Co. Ltd. [4]Peng Cheng Laboratory
eecyw@mail.scut.edu.cn, kuijia@scut.edu.cn

## Abstract

Creation of 3D content by stylization is a promising yet challenging problem in computer vision and graphics research. In this work, we focus on stylizing photorealistic appearance renderings of a given surface mesh of arbitrary topology. Motivated by the recent surge of cross-modal supervision of the Contrastive Language-Image Pre-training (CLIP) model, we propose TANGO, which transfers the appearance style of a given 3D shape according to a text prompt in a photorealistic manner. Technically, we propose to disentangle the appearance style as the spatially varying bidirectional reflectance distribution function, the local geometric variation, and the lighting condition, which are jointly optimized, via supervision of the CLIP loss, by a spherical Gaussians based differentiable renderer. As such, TANGO enables photorealistic 3D style transfer by automatically predicting reflectance effects even for bare, low-quality meshes, without training on a task-specific dataset. Extensive experiments show that TANGO outperforms existing methods of text-driven 3D style transfer in terms of photorealistic quality, consistency of 3D geometry, and robustness when stylizing low-quality meshes. Our codes and results are available at our project webpage https://cyw-3d.github.io/tango.

## 1 Introduction

3D content creation by stylization (e.g., stylized according to text prompts [31], images [44], or 3D shapes [50]) has important applications in computer vision and graphics areas. The problem is yet challenging and traditionally requires manual efforts from experts of professional artists and a large amount of time cost. In the meanwhile, many online 3D repositories [6, 47, 41] can be easily accessed on the Internet whose contained surface meshes are bare contents without any styles. It is thus promising if automatic, diverse, and realistic stylization can be achieved given these raw 3D contents. We note that similar to recent 3D stylization works [31, 50, 18, 33, 21], we consider *style* as the particular appearance of an object, which is determined by color (albedo) and physical reflective effect of the object surface, while considering *content* as the global shape structure and topology prescribed by an explicit 3D mesh or other implicit representation of the object surface.

Stylization is usually guided by some sources of styling signals, such as a text prompt [31], a reference image [44], or a target 3D shape [50]. In this work, we choose to work with stylization by text prompts, motivated by the surprising effects recently achieved in many applications [44, 19, 10], by using the cross-modal supervision model of Contrastive Language–Image Pre-training (CLIP) [37]. The goal of this paper is to devise an end-to-end neural architecture system, named TANGO, that can transfer, guided by a text prompt, the style of a given 3D shape of arbitrary topology. Note that TANGO can be directly applied to arbitrary meshes with arbitrary target styles, without additional learning/optimization procedures on a task-specific dataset; Figure 1 gives the illustration.

---

*Corresponding Author

36th Conference on Neural Information Processing Systems (NeurIPS 2022).

The most relevant work so far is Text2Mesh [31], which is the first to use the CLIP loss in mesh stylization; given an input mesh and a text prompt, it predicts stylized color and displacement for each mesh vertex, and the stylized mesh is just a result of this colored vertex displacement procedure. However, Text2Mesh does not support the stylization of lighting conditions, reflectance properties, and local geometric variations, which are necessary in order to produce a photorealistic appearance of 3D surface. To this end, we formulate the problem of mesh stylization as learning three unknowns of the surface, i.e., the spatially varying Bidirectional Reflectance Distribution Function (SVBRDF), the local geometric variation (normal map), and the lighting condition. We show that TANGO is able to learn the above unknowns supervised only by CLIP-bridged text prompts, and generate photorealistic rendering effects by approximating these unknowns during inference. Technically, by jointly encoding the text prompt and images of the given mesh rendered by a spherical Gaussians based differentiable renderer, TANGO is able to compare the embeddings of text and mesh style and then backpropagate the gradient signals to update the style parameters. Our approach disentangles the style into three components represented by learnable neural networks, namely continuous functions respectively of the point, its normal, and the viewing direction on the surface. Note that due to the learning of neural normal map, TANGO can produce fine-grained geometric details even on low-quality meshes with only a few faces; in contrast, vertex displacement-based methods, such as [31], often fail in such cases.

We finally summarize our contributions as follows.

- We propose a novel, end-to-end architecture (TANGO) for *text-driven photorealistic 3D style transfer*; our model learns to assign advanced physical lighting appearance and local geometric details to a raw mesh, by leveraging an off-the-shelf, pre-trained CLIP model.

- TANGO can automatically predict material, normal map, and lighting condition for any bare mesh prescribed by text description, and can also handle low-quality meshes with only a few faces, due to our prediction of colors and reflectance properties for every intersection point of a camera ray.

- We conduct extensive ablation studies and experiments that show the advantages of TANGO. Notably, except for more photorealistic renderings, our results have no self-intersection in local geometry details; in contrast, such imperfections would appear in results from existing methods.

## 2 Related Work

**Text-Driven Generation and Manipulation.** Our work is primarily inspired by [31] and other methods [44, 19, 40, 20, 22] which use text prompt to guide 3D generation or manipulation. The optimization procedures of these methods are guided by the CLIP model [37]. Specifically, CLIP-NeRF [44] proposes a unified framework that allows manipulating NeRF, guided by either a short text prompt or an examplar image. Another work Dreamfields [19] chooses to generate 3D scenes from scratch in a NeRF representation. Different from those NeRF-based methods, CLIP-Forge [40] presents a method for shape generation represented by voxels. Text2Mesh [31] predicts color and geometric details for a given template mesh according to the text prompt. Concurrently, Khalid *et al.* [22] generate stylized mesh from a sphere guided by CLIP. Meanwhile, apart from 3D content creation, there are other works focusing on text-guided image generation and manipulation. StyleCLIP [36] and VQGAN-CLIP [10] use CLIP-loss to optimize latent codes in GAN's style space. GLIDE [35] is introduced for more photorealistic image generation and editing with text-guided diffusion models, while DALL·E [38] explores to generate images using transformer. Compared to these methods, we focus on improving the realism of text-driven 3D mesh stylization.

**3D Style Transfer.** Generating or editing 3D content according to a given style is a long-standing task in computer graphics [12, 14, 48]. NeuralCage [49] implements traditional cage-based deformation as differentiable network layers and generates feature-deserving deformations. Furthermore, 3DStyleNet [50] conducts 3D object stylization informed by a reference textured 3D shape. Other works are specific to styles of furniture [27], 3D colleges [12], LEGO [23], and portraits [17]. Compared to these global geometric transfer approaches, some works investigate style transfer on a more local level. Specifically, Hertz *et al.* [18] propose a generative framework to learn local structures from the style of a given mesh, while Liu *et al.* [25] subdivide the given mesh for a coarse-to-fine

geometric modeling. Unlike these methods, we aim to synthesize a wide range of realistic styles specified by a text prompt, making it more convenient to guide the process by high-level semantics.

**BRDF and Lighting Estimation.** Previously, BRDF and lighting can only be estimated under complicated settings in a laboratory environment [2, 28, 46] before the deep learning era. With the help of neural networks, some techniques can use simpler settings to estimate BRDF and lighting beyond geometry [13, 3, 1, 11]. They assume that scenes are under specific illumination, like one [11] or multiple [13] flashlights and linear light [15]. However, they often assume that the geometry is a plane [13] or known [3]. These requirements are difficult to be met in reality. Furthermore, recent works [7, 8, 26, 53] exploit neural networks to jointly predict geometry, BRDF and in some cases, lighting via 2D image loss. These methods can be split into two categories according to the geometry representation. Among them, those who use explicit representation [26, 7, 8] usually deform a sphere to obtain final shapes, which cannot represent arbitrary topology. Another pipeline of methods that use implicit geometry representation like SDF [51], occupancy function [30] or NeRF [42, 4, 52] can get higher quality shapes, but these representations are not handy to be processed by contemporary game engines. Unlike them, our task is to change the style of a given shape, so the explicit triangle mesh is preferred due to its convenience. As for BRDF and lighting representation, PhySG [51] and NeRD [4] use mixtures of spherical Gaussians to represent illumination; NeRFactor [52] and nvdiffrec [34] choose low-resolution envmaps and split sum lighting model, respectively. Different from all these 2D supervision methods, we jointly estimate spatially varying BRDF, lighting, and local geometry (normal map) supervised by text prompts. With the help of estimated BRDF and lighting, we can represent complicated light reflection in the real world on stylized meshes.

## 3   Method

We contribute TANGO, an end-to-end architecture that enables *text-driven photorealistic 3D style transfer* for any bare mesh, which is supervised by a semantic loss. The heart of TANGO is to disentangle the style of the input mesh as reflectance properties and scene lighting. Then, given a target style specified by a text prompt, we could learn the corresponding style parameters by leveraging the pre-trained CLIP model, and then the stylized images could be generated with the learned style parameters through our spherical Gaussians (SG) differentiable renderer. There are three unknowns in our defined style parameters: (i) *spatially varying BRDF (SVBRDF)*, including diffuse, roughness, specular, represented by parameters $\boldsymbol{\xi} \in \mathbb{R}^m$; (ii) *normal*, represented by parameters $\boldsymbol{\gamma} \in \mathbb{R}^n$; (iii) *lighting*, represented by parameters $\boldsymbol{\tau} \in \mathbb{R}^l$.

We use explicit triangle mesh representation to represent the input 3D shape. The input template mesh $M$ consists of $e$ vertices $\boldsymbol{V} \in \mathbb{R}^{e \times 3}$ and $u$ faces $\boldsymbol{F} \in \{1, ..., n\}^{u \times 3}$ and is fixed throughout training. The object's style is described by a text prompt, and the aforementioned SVBRDF, normal and lighting parameters are optimized to fit the description. Compared to other implicit representation [51, 4], the explicit representation here is more convenient to obtain and easy to use by current game engines and other applications.

**Forward model.**   Given an object mesh $M$, we first scale it in a unit sphere and then randomly sample points near $M$ as camera positions $\boldsymbol{c}$ to render images. For each pixel in rendered images, indexed by $p$, let $\boldsymbol{R}_p = \{\boldsymbol{c}_p + t\boldsymbol{\nu}_p \mid t \geq 0\}$ denotes the ray through pixel $p$, where $\boldsymbol{c}_p$ is randomly sampled on a sphere containing the object and $\boldsymbol{\nu}_p$ denotes the direction of the ray (i.e. the vector pointing from $\boldsymbol{c}_p$ to $p$). Let $\boldsymbol{x}_p$ denotes the first intersection of the ray $\boldsymbol{R}_p$ and the mesh $M$, $\boldsymbol{n}_p$ denotes the ground truth normal of point $\boldsymbol{x}_p$, $\hat{\boldsymbol{n}}_p = \boldsymbol{\Pi}(\boldsymbol{n}_p, \boldsymbol{x}_p; \boldsymbol{\gamma})$ is the estimated normal on surface point $\boldsymbol{x}_p$. For each surface point $\boldsymbol{x}_p$ with the estimated normal $\hat{\boldsymbol{n}}_p$, we suppose that $\boldsymbol{L}_i(\boldsymbol{\omega}_i; \boldsymbol{\tau})$ is the incident light intensity from direction $\boldsymbol{\omega}_i$, and SVBRDF $\boldsymbol{f}_r(\boldsymbol{\nu}_p, \boldsymbol{\omega}_i, \boldsymbol{x}_p; \boldsymbol{\xi})$ are the surface reflectance coefficients of the material at location $\boldsymbol{x}_p$ from viewing direction $\boldsymbol{\nu}_p$ and incident light direction $\boldsymbol{\omega}_i$. Then the observed light intensity $\boldsymbol{L}_p(\boldsymbol{\nu}_p, \boldsymbol{x}_p, \boldsymbol{n}_p)$ is an integral over the hemisphere $\Omega = \{\boldsymbol{\omega}_i : \boldsymbol{\omega}_i \cdot \hat{\boldsymbol{n}}_p \geq 0\}$:

$$\boldsymbol{L}_p(\boldsymbol{\nu}_p, \boldsymbol{x}_p, \boldsymbol{n}_p; \boldsymbol{\xi}, \boldsymbol{\gamma}, \boldsymbol{\tau}) = \int_{\Omega} \boldsymbol{L}_i(\boldsymbol{\omega}_i; \boldsymbol{\tau}) \boldsymbol{f}_r(\boldsymbol{\nu}_p, \boldsymbol{\omega}_i, \boldsymbol{x}_p; \boldsymbol{\xi})(\boldsymbol{\omega}_i \cdot \boldsymbol{\Pi}(\boldsymbol{n}_p, \boldsymbol{x}_p; \boldsymbol{\gamma})) \mathrm{d}\boldsymbol{\omega}_i. \qquad (1)$$

For each image $I \in [0, 1]^{H \times W \times 3}$ corresponding to a camera position $\boldsymbol{c}$, we calculate the pixel color by Equation (1). The rendered image is encoded into a latent code by a pre-trained CLIP model,

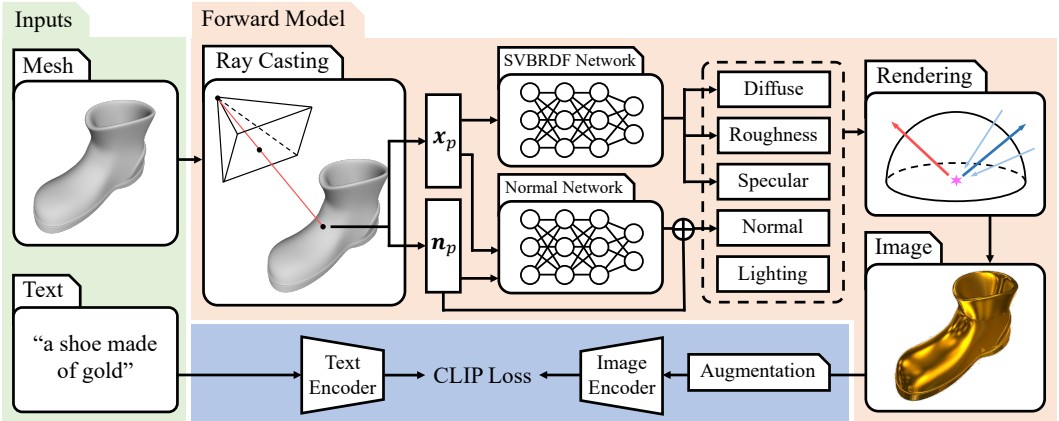

Figure 1: Given a scaled bare mesh and a text prompt (e.g., "a shoe made of gold" as in this figure), we first specify a camera location $c$ and cast rays $\boldsymbol{R}$ to intersect with the given mesh. For each intersecting camera ray, we obtain a surface point $\boldsymbol{x}_p$ and normal $\boldsymbol{n}_p$, which are used for predicting spatially varying BRDF parameters and normal variation by two MLPs, i.e., the SVBRDF Network and Normal Network. Meanwhile, the lighting is represented using a series of spherical Gaussians parameters $\{\boldsymbol{\mu_k}, \lambda_k, \boldsymbol{a}_k\}$. For each iteration, we render the image using a differentiable SG renderer and then encode the augmented image using the image encoder of a CLIP model, which backpropagates the gradients to update all learnable parameters.

which is expected to be aligned with the encoded text prompt. Finally, a loss measuring the divergence between encoded image and text representations is used to simultaneously train the model parameters, i.e., $\boldsymbol{\xi}, \boldsymbol{\gamma}$, and $\boldsymbol{\tau}$, resulting in photorealistic renderings matching the text description. We summarize our approach in Figure 1 and clarify each step in the following: Sec. 3.1 outlines our intersection calculation, Sec. 3.2 presents our appearance style model, Sec. 3.3 illustrates the rendering procedure, and the loss function we use to train our network is described in Sec. 3.4.

## 3.1  Intersection of camera rays and mesh

Given an input 3D mesh $M$ and target text, we first scale $M$ in a unit sphere and find an anchor view on the sphere, then other camera positions are randomly sampled using Gaussian distribution centered around the anchor view, whose direction vectors are set to look at the origin of the unit bounding sphere. Then for each sampled camera position $c$ and a pixel $p$, we get a camera ray $\boldsymbol{R}_p = \{\boldsymbol{c} + t\boldsymbol{\nu}_p \mid t \geq 0\}$, pointing from $c$ to $p$. After that, we use ray casting [39] to find the first intersection point $\boldsymbol{x}_p$ and intersection face $\boldsymbol{f}_p$ of $\boldsymbol{R}_p$ and mesh $M$, while the face normal $\boldsymbol{n}_p$ of $\boldsymbol{f}_p$ is used as surface normal at point $\boldsymbol{x}_p$.

## 3.2  Appearance modeling

To generate a photorealistic style of object $M$, we use three components in Equation (1) to model appearance: (1) an environment map, (2) a normal map, and (3) SVBRDF consists of diffuse term and specular term. To calculate Equation (1), some works use Monte Carlo sampling [24], split sum approximation [34], spherical Gaussians [51] or spherical Harmonics [16] to solve the integration. In this paper, we choose to use spherical Gaussians (SG) to efficiently approximate the rendering equation in closed form.

An $n$-dimensional spherical Gaussian is a spherical function that takes the following form [45]:

$$G(\boldsymbol{\nu}; \boldsymbol{\mu}, \lambda, \boldsymbol{a}) = \boldsymbol{a}e^{\lambda(\boldsymbol{\nu}\cdot\boldsymbol{\mu}-1)}, \tag{2}$$

where $\boldsymbol{\nu} \in \mathbb{S}^2$ is the function input, $\boldsymbol{\mu} \in \mathbb{S}^2$ is the lobe axis, $\lambda \in \mathbb{R}_+$ is the lobe sharpness, and $\boldsymbol{a} \in \mathbb{R}_+^n$ is the lobe amplitude ($\boldsymbol{a} \in \mathbb{R}_+^3$ for RGB color). Spherical Gaussians have two nice properties to ensure the convenience of calculating integration:

(1) the multiplication of two spherical Gaussians is another spherical Gaussian:

$$G(\boldsymbol{\nu}; \boldsymbol{\mu_1}, \lambda_1, \boldsymbol{a_1})G(\boldsymbol{\nu}; \boldsymbol{\mu_2}, \lambda_2, \boldsymbol{a_2}) = G(\boldsymbol{\nu}; \frac{\boldsymbol{\mu_m}}{\|\boldsymbol{\mu_m}\|}, (\lambda_1+\lambda_2)\|\boldsymbol{\mu_m}\|, \boldsymbol{a_1}\boldsymbol{a_2}e^{\lambda_m(\|\boldsymbol{\mu_m}\|-1)}), \quad (3)$$

where $\lambda_m = \lambda_1 + \lambda_2$, $\boldsymbol{\mu_m} = \frac{\lambda_1\boldsymbol{\mu_1}+\lambda_2\boldsymbol{\mu_2}}{\lambda_1+\lambda_2}$.

(2) the integration of a single spherical Gaussian has a closed-form solution:

$$\int_\Omega G(\boldsymbol{\nu}; \boldsymbol{\mu}, \lambda, \boldsymbol{a})\mathrm{d}\boldsymbol{\nu} = 2\pi\frac{\boldsymbol{a}}{\lambda}(1 - e^{-2\lambda}). \quad (4)$$

The above two properties guarantee that if we use SG to represent each multiplication term in Equation (1), the formula in integral can be written as a single SG, which can be calculated efficiently.

**Environment map.** We represent environment map $\boldsymbol{L}_i(\boldsymbol{\omega}_i)$ with multiple spherical Gaussians (SG):

$$\boldsymbol{L}_i(\boldsymbol{\omega}_i) = \sum_{k=0}^{M} G(\boldsymbol{\omega}_i; \boldsymbol{\mu_k}, \lambda_k, \boldsymbol{a}_k). \quad (5)$$

The light energy is:

$$E(\boldsymbol{L}) = \sum_{k=0}^{M} \frac{2\pi\boldsymbol{a}_k}{\lambda_k(1 - e^{-2\lambda_k})}. \quad (6)$$

When energy is too large or too small, the rendered image will be too bright or too dark, resulting in the optimization falling to a local minimum. To alleviate this issue, we initialize the total energy to 6.25.

**Normal map.** To provide more geometry details for meshes when rendering, for each $\boldsymbol{x}_p$ and $\boldsymbol{n}_p \in \{(1, \theta_p, \varphi_p)|\theta_p \in (0, 2\pi), \varphi_p \in (0, \pi)\}$, we estimate a normal offset to $\boldsymbol{n}_p$ to get estimated normal $\hat{\boldsymbol{n}}_p \in \left\{(1, \hat{\theta}_p, \hat{\varphi}_p)|\hat{\theta}_p \in (0, 2\pi), \hat{\varphi}_p \in (0, \pi)\right\}$:

$$(\hat{\theta}_p, \hat{\varphi}_p) = (\theta_p + \triangle\theta, \varphi_p + \triangle\varphi) = \boldsymbol{\Pi}(\beta(\boldsymbol{x}_p, \theta_p, \varphi_p); \boldsymbol{\gamma}), \quad (7)$$

$$(\triangle\theta, \triangle\varphi) = \triangle\boldsymbol{\Pi}(\beta(\boldsymbol{x}_p, \theta_p, \varphi_p); \boldsymbol{\gamma}), \quad (8)$$

where $\triangle\boldsymbol{\Pi}$ is a neural network (MLP) estimating the normal offset. For every input, we apply a positional encoding layer $\beta(l) = [\cos(2\pi\boldsymbol{B}l), \sin(2\pi\boldsymbol{B}l)]^T$ to synthesize high-frequency details, where $\boldsymbol{B}$ is a random Gaussian matrix whose entry is randomly drawn from $\mathcal{N}(0, \sigma^2)$ or an arithmetic sequence ranging from 1 to N like NeRF [32]. $(\hat{\theta}_p, \hat{\varphi}_p)$ is clamped to lie in reasonable ranges.

**SVBRDF.** Below we take apart the SVBRDF $\boldsymbol{f}_r(\boldsymbol{\nu}, \boldsymbol{\omega}_i, \boldsymbol{x}_p)$ into diffuse term $\boldsymbol{f}_d(\boldsymbol{x}_p)$ and specular term $\boldsymbol{f}_s(\boldsymbol{\nu}, \boldsymbol{\omega}_i, \boldsymbol{x}_p)$. For diffuse term, we use an MLP $\boldsymbol{\Phi}_1(\beta(\boldsymbol{x}_p); \boldsymbol{\xi}_1)$ to calculate spatially varying diffuse albedo, where $\beta(l)$ is positional encoding layer. Then diffuse term is

$$\boldsymbol{f}_d(\boldsymbol{x}_p) = \frac{\boldsymbol{\Phi}_1(\beta(\boldsymbol{x}_p); \boldsymbol{\xi}_1)}{\pi}. \quad (9)$$

For the specular term, we use the simplified Disney BRDF model [5]:

$$\boldsymbol{f}_s(\boldsymbol{\nu}, \boldsymbol{\omega}_i) = \mathcal{M}(\boldsymbol{\nu}, \boldsymbol{\omega}_i)\mathcal{D}(\boldsymbol{\nu}, \boldsymbol{\omega}_i), \quad (10)$$

where $\mathcal{M}$ accounts for Fresnel and shadowing effects, and $\mathcal{D}$ is normal distribution. Following [51], we use spherical Gaussians to represent specular term, so the specular SVBRDF function can be written as:

$$\boldsymbol{f}_s(\boldsymbol{\nu}, \boldsymbol{\omega}_i, \boldsymbol{x}_p) = \sum_{j=0}^{N} G(\boldsymbol{h}; \hat{\boldsymbol{n}}_p, \frac{\lambda_j}{4\boldsymbol{h} \cdot \boldsymbol{\nu}}, \mathcal{M}_p\boldsymbol{a}_j), \quad (11)$$

where the half vector $\boldsymbol{h} = \frac{\boldsymbol{\nu}+\boldsymbol{\omega}_i}{\|\boldsymbol{\nu}+\boldsymbol{\omega}_i\|_2}$, the approximated Fresnel and shadowing term $\mathcal{M}_p \approx \mathcal{M}(\boldsymbol{\nu}, 2(\boldsymbol{\nu}\cdot\hat{\boldsymbol{n}}_p)\hat{\boldsymbol{n}}_p - \boldsymbol{\nu})$. $\{\lambda_j, \boldsymbol{a}_j\}$ are predicted by an MLP $\boldsymbol{\Phi}_2(\beta(\boldsymbol{x}_p); \boldsymbol{\xi}_2)$. The complete SVBRDF function is

$$\boldsymbol{f}_r(\boldsymbol{\nu}, \boldsymbol{\omega}_i, \boldsymbol{x}_p) = \boldsymbol{f}_d(\boldsymbol{x}_p) + \boldsymbol{f}_s(\boldsymbol{\nu}, \boldsymbol{\omega}_i, \boldsymbol{x}_p). \quad (12)$$

Following [29], we use an SG to approximate term $\boldsymbol{\nu} \cdot \hat{\boldsymbol{n}}_p = G(\boldsymbol{\nu}; 0.0315, \hat{\boldsymbol{n}}_p, 32.7080) - 31.7003$. Note that all components in Equation (1) are represented by SGs, so the integration can be calculated in closed form.

### 3.3 Rendering

Given our geometric and appearance components, we perform rendering of a ray $R_p$'s color as follows: (1) find the intersection point $x_p$ and normal $n_p$ between $R_p$ and the mesh using ray casting; (2) predict the normal offset and get the estimated normal $\hat{n}_p$; (3) compute the diffuse albedo $\Phi_1(x_p; \xi_1)$ and specular albedo $\Phi_2(x_p; \xi_2)$; (4) use estimated normal $\hat{n}_p$, environment map $\{\mu_k, \lambda_k, a_k\}$, SVBRDF $\Phi(x_p; \xi)$, and viewing direction $\nu_p$ to compute the color for $R_p$; (5) conduct steps (1)-(4) repeatedly for every pixel and then finally gather them as the rendered image. The overall pipeline is illustrated in Figure 1. Note that our pipeline is fully differentiable w.r.t. all the optimizable parameters.

### 3.4 Text-based correspondence loss

Our optimization is guided by a pre-trained CLIP [37] model. For each iteration, we sample a batch of camera positions. For each camera position $c$, we can render an image $I \in [0, 1]^{H \times W \times 3}$ whose pixel's color is calculated by Equation (1) and a mask $O \in \{0, 1\}^{H \times W}$. Then we use the mask $O$ to add background to image $I$. We test black, white, and random Gaussian backgrounds, and observe similar stylized performance. Therefore, we use white and black backgrounds in TANGO for convenience. Next, in the 2D augmentation phase, we randomly crop some areas of $I$ and resized them to (224, 224), resulting in the augmentations of $I_a$. Subsequently, $I$ and augmented images $I_a$ are all encoded into the multimodal space by the image encoder of CLIP, generating an averaged latent code $\mathcal{L}_i \in \mathbb{R}^{512}$. Meanwhile, the input text prompt is encoded to $\mathcal{L}_t \in \mathbb{R}^{512}$ by the text encoder of CLIP. Finally, we use the following cosine similarity as the overall objective:

$$Loss(\mathcal{L}_i, \mathcal{L}_t) = -\frac{\mathcal{L}_i \cdot \mathcal{L}_t}{\|\mathcal{L}_i\|_2 \|\mathcal{L}_t\|_2}. \tag{13}$$

**Implementation details.** The environment map is represented by 32 or 64 spherical Gaussians, and the specular BRDF is represented by 1 SG. The normal estimation network $\Pi(x_p, \theta_p, \varphi_p; \gamma) \in \mathbb{R}^2$ consists of 3 layers with hidden layers of width 256. The SVBRDF network, $\Phi(x_p; \xi) \in \mathbb{R}^7$, with hidden layers of width 256, predicts diffuse, specular and roughness per surface point; the three ones share the first 2 layers of $\Phi(x_p; \xi)$ and the parameters in next 3 layers are exclusive. We use the pre-trained CLIP model with the ViT-B/32 backbone. We adopt the AdamW optimizer and the initial learning rate is set as $5 \times 10^{-4}$. The learning rate is decayed by 0.7 in every 500 iterations.

## 4 Experiment

Following Text2Mesh[31], we examine TANGO in a variety of meshes and different text prompts. The object meshes are collected from: COSEG[41], Thingi10K[54], ShapeNet[6], Turbo Squid[43], and ModelNet[47]. To test TANGO's robustness to low-quality meshes, we downsample the number of meshes' vertices and faces in Sec. 4.2, while in other experiments we directly stylize the given mesh without any pre-processing. Except in Sec. 4.2, meshes used in our paper contain an average of $73,966$ faces, $16\%$ non-manifold edges, $0.2\%$ non-manifold vertices, and $12\%$ boundaries. Before stylizing the given mesh, we scale the input into a unit sphere and move the center of its vertices to the origin. TANGO is trained on a single Nvidia RTX 3090 GPU and takes about 10 minutes for 1500 iterations.

In Sec. 4.1, we examine TANGO on different meshes with text prompts and compare our results to state-of-the-art text-driven mesh stylization approach Text2Mesh[31]. After that, in Sec. 4.2, we degenerate the quality of meshes by downsampling the number of faces to 5000, verifying the robustness of TANGO to sparse mesh data. Next in Sec. 4.3, we conduct the ablation study to explore the impact of our proposed modules.

### 4.1 Neural stylization and controls

TANGO generates photorealistic details with fine granularity and avoids self-intersection in the meantime. As presented in Figure 2, TANGO not only presents realistic reflective effects like highlight and shade in smooth materials like gold and silver but also tackles uneven surfaces by producing dark and bright shadings via the estimated normals. Furthermore, in Figure 3, our approach

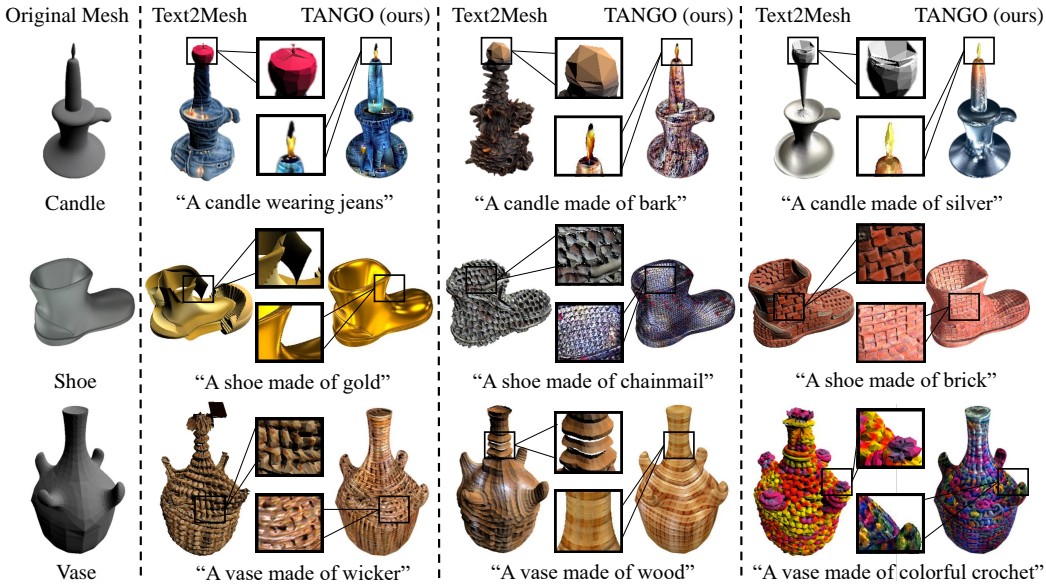

Figure 2: 3D stylization results of TANGO and Text2Mesh given the same mesh and text prompt. TANGO produces results of higher fidelity and more realistic rendering. See Sec. 4.1 for the details.

Table 1: Mean opinion scores (1-5) for Q1-Q3 (cf. Sec. 4.1), for TANGO and baseline.

|  | (Q1): Overall | (Q2): Content | (Q3): Style |
|---|---|---|---|
| Text2Mesh | $3.30(\pm0.75)$ | $3.53(\pm0.79)$ | $3.42(\pm0.66)$ |
| Ours | $\mathbf{4.02}(\pm0.77)$ | $\mathbf{3.98}(\pm0.74)$ | $\mathbf{3.94}(\pm0.75)$ |

demonstrates a consistent 3D stylization in different views and exhibits natural variation in texture. Similar to Text2Mesh, we incorporate 3D symmetry prior for those meshes that need style symmetry across an axis, like the person shown in Figure 3. Specifically, for each point $\boldsymbol{x}_p = (x, y, z)$ and a shape with bilateral symmetry across the X-Y plane, we apply a symmetry prior to the positional encoder, i.e., replacing $\beta(x, y, z)$ as $\beta(x, y, |z|)$. With this symmetry prior, the stylized Iron Man in Figure 3 is symmetrical across the z-axis, which satisfies the real situation for a person. It is surprising that we can generate such photorealistic renderings only supervised by CLIP loss. Besides the powerful pre-trained CLIP model, the key point of our success is the shading model and carefully designed disentangled elements in Equation (1). By predicting normal offsets and SVBRDF individually, optimizing the light SG parameters, and inputting these elements into the physical rendering equation, the individual module predicts physics-aware results, as illustrated with the disentangled components in Figure 4. Meanwhile, because TANGO disentangles geometry, material and lighting representation, we can easily relight the stylized mesh, which can not be accomplished by vertex displacement-based methods. In Figure 3, we conduct the relighting experiments with the example of "A vase made of wood". By replacing the estimated environment map with others downloaded from the Internet, the lighting of the rendering results could be successfully manipulated. Another benefit of our disentangled mesh style representation is that we can easily edit the physical elements of materials to get the preferred style, which is important for incorporating current graphic engines. As Figure 6 shows, the surface of the golden shoe becomes more diffuse as we enlarge the roughness value of the material, while a larger specular value typically results in a more shiny surface, proving the capability of TANGO on material editing.

**Comparison with Text2Mesh.** In this part we compare TANGO with a pioneering work in text-driven 3D stylization, Text2Mesh [31], which operates on the point colors and vertex's displacement. We use the official implementation of Text2Mesh and also train it for 1500 iterations for a fair comparison. Compared to Text2Mesh, TANGO can handle both smooth mental appearance and rough materials, generating realistic renderings with highlight effects. For example in Figure 2, given

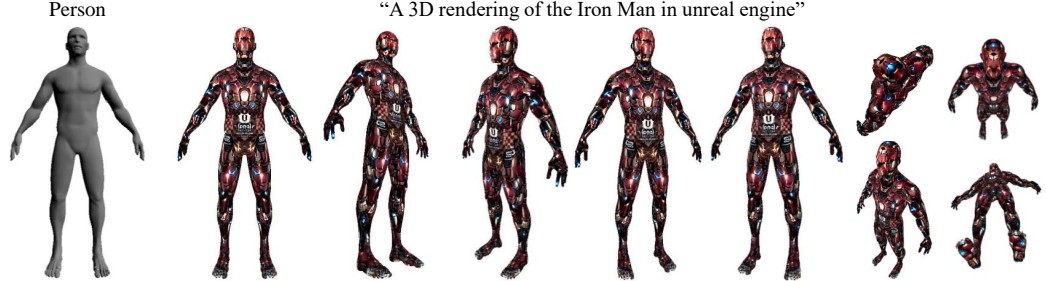

Person            "A 3D rendering of the Iron Man in unreal engine"

Figure 3: Given a bare person mesh and text prompt, TANGO can stylize the entire 3D Iron Man appearance and therefore generates 3D-consistent renderings.

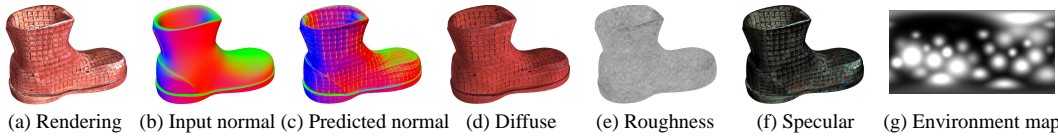

(a) Rendering   (b) Input normal   (c) Predicted normal   (d) Diffuse    (e) Roughness     (f) Specular    (g) Environment map

Figure 4: An illustration of the disentangled rendering components. This figure shows (a) our final rendering result of "A shoe made of brick" and individual components for the rendering including (b) the original normal map of input mesh, (c) the normal map predicted by TANGO, (d) diffuse map predicted by TANGO, (e) spatially varying roughness map, (f) spatially varying specular map and (g) environment map. Note that the BRDF is composed of the diffuse map, roughness map, and specular map.

a bare shoe mesh and a text prompt "a shoe made of gold", TANGO can produce realistic golden reflectance that has light and shade contrast, which is not observed in the competitive Text2Mesh. For materials that need an apparent bump, for example, "a vase made of wicker", our approach successfully generates bump materials without incorporating local self-intersection. In addition, we randomly chose 73 users to evaluate 9 source meshes and style text prompt combinations. Each of them was asked three questions: (Q1) "How natural is the output results?" (Q2) "How well does the output match the original content?" (Q3) "How well does the output match the target style?" We report the mean opinion scores with standard deviations for both Text2Mesh and ours in Table 1. TANGO outperforms the Text2Mesh baseline across all questions, with an advantage of 0.72, 0.45, and 0.52 for Q1-Q3, respectively. More in-depth discussions and comparisons with Text2Mesh appear in our appendices.

## 4.2 Robustness to low-quality meshes

To measure the robustness of our approach to low-quality meshes, we show stylization results of original meshes and downsampled meshes in Figure 7. The original lamp mesh and alien mesh have $41,160$ faces and $68,430$ faces, respectively. To generate the low-quality meshes, we both

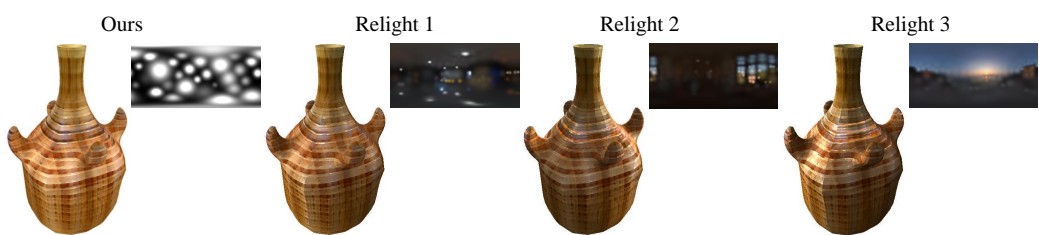

Ours            Relight 1            Relight 2            Relight 3

Figure 5: An illustration of the relighting performance. The left image shows our rendering result and estimated environment map of "A vase made of wood", while the following three ones are relighted results with different environment maps downloaded from the Internet.

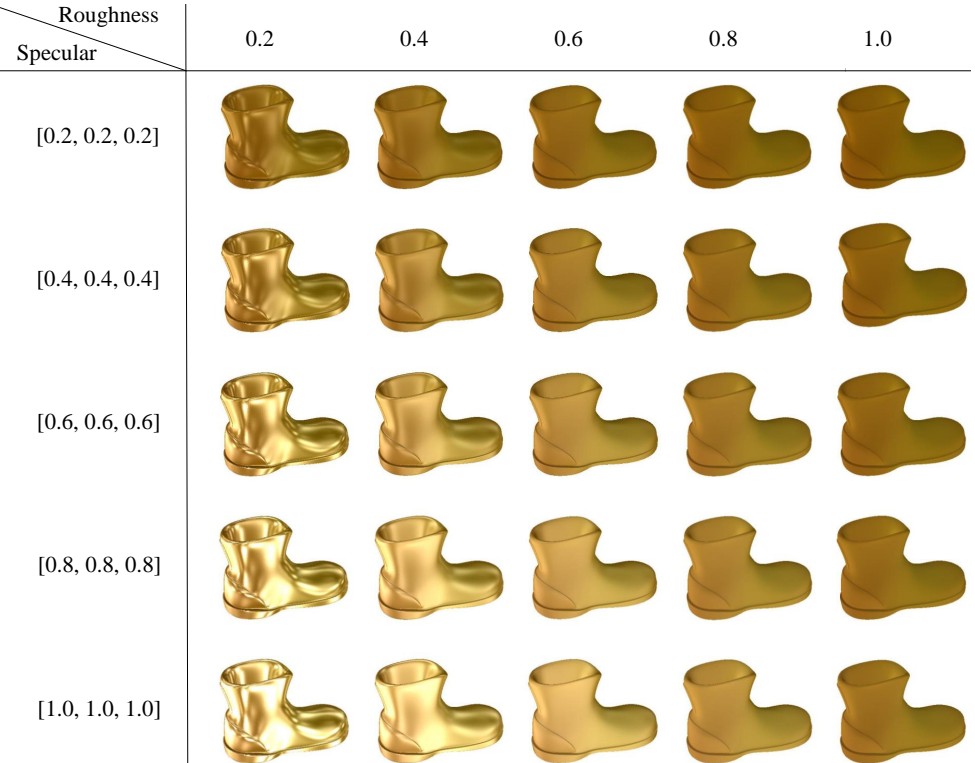

Figure 6: An illustration of the material editing performance. In each line, we increase the roughness value of the material, while we increase the specular value in each column. We can observe that the surface becomes more diffuse as we enlarge the roughness, while increasing the specular value results in more shiny surfaces. This illustration justifies the capability of TANGO on editing the material.

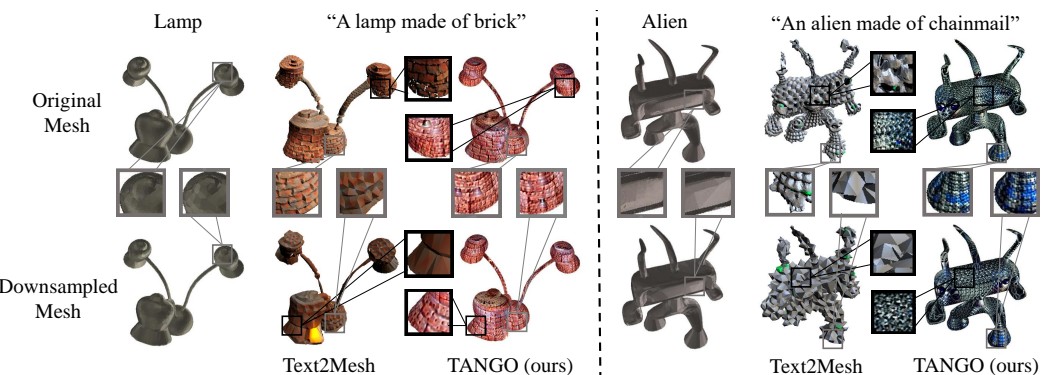

Figure 7: A robustness comparison of TANGO and Text2Mesh on high-quality meshes (cf. the top row) and downsampled meshes (cf. the second row). The results of Text2Mesh degenerate significantly as the mesh quality downgrades, while TANGO presents consistent results on meshes of various qualities.

downsample them to $5,000$ faces using Meshlab [9]. On low-quality meshes with fewer faces, TANGO still maintains a similar performance to the original mesh, while Text2Mesh is trapped in self-intersection and monotonous color prediction. The reason for the phenomenon is that we make a spatially varying prediction for each intersection point of the camera ray and mesh, and the local geometric variation compensates for the lack of geometric details in low-quality meshes. As for displacement-based methods like Text2Mesh, they rely on moving each vertex's position to generate bumpy geometry, which makes them deviate with vertices number shrinking.

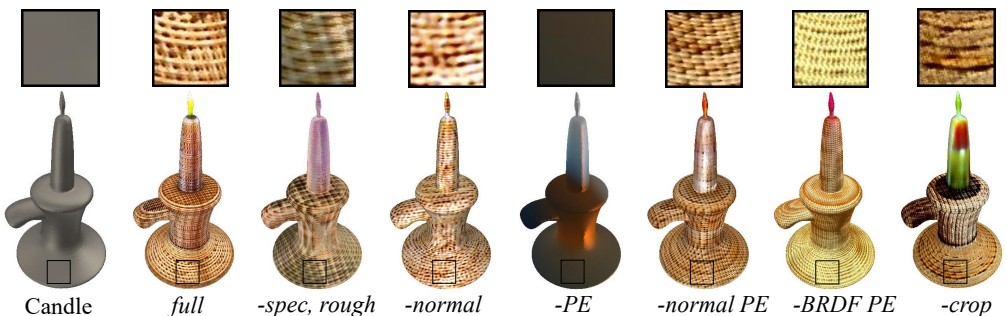

Figure 8: Ablation experiments on the proposed modules in TANGO, where the text prompt is "A candle made of wicker". Please refer to the Sec. 4.3 for the detailed definitions.

## 4.3 Ablation study

To verify the effectiveness of each module in our approach, we conduct an ablation study by removing different components from TANGO. The results are shown in Figure 8. Compared to the full model, when we remove spatially varying specular and roughness prediction (*-spec, rough* in Figure 8), the whole SVBRDF is only represented by diffuse color and therefore it cannot generate realistic view-dependent highlight in renderings. If we disable the normal estimation network (*-normal* in Figure 8), the stylized appearance becomes smooth and fails to generate enough geometric variations, verifying the effectiveness of the normal estimation network in generating geometric bump details. To provide enough high-frequency details when stylizing, we use the positional encoding layer to wrap the input points and normals for both SVBRDF prediction and the normal estimation network. If we remove positional encoder (*-PE*, *-normal PE* and *-BRDF PE* in Figure 8), our results will lack in high-frequency variation in both color and geometry. In addition, local crop augmentation (*-crop* in Figure 8) also contributes obviously to the stylized performance, since it makes our network focus on small areas on the surface and avoid blurred results.

## 5 Limitations and conclusions

Our main limitation is the simplified shading model to represent indirect lighting effects and shadow, impeding TANGO from representing complicated lighting in large scenes which need to simulate multiple light bounces. The reason for choosing spherical Gaussians to represent our shading is to accelerate the rendering and training process. To extend our model for large scene stylization, indirect lighting term and acceleration techniques can be incorporated to guarantee both quality and speed. Another limitation is that the vertex displacement framework presents a larger geometry capacity than ours in theory. However, the rendering capacity gap between frameworks of vertex displacement and our adopted normal displacement could be largely reduced with our introduced learnable SVBRDF and normal, meanwhile presenting more robustness and time efficiency. TANGO and the recent Text2Mesh conduct the initial attempts to this problem in different directions, which may inspire more in-depth following investigations.

In this paper, we propose TANGO, a novel framework to generate photorealistic appearance style for an arbitrary 3D mesh according to a text prompt. By disentangling the appearance style as SVBRDF, local geometric variation, and lighting condition, we jointly learn them under the supervision of the CLIP loss via a spherical Gaussians based differentiable renderer. Compared to existing competitors, TANGO could automatically predict reflectance effects even for bare, low-quality meshes. However, on the other hand, it will possibly make junior artists responsible for creating simple texture and lighting for bare meshes out of employment.

## Acknowledgements

This work is supported in part by Guangdong R&D key project of China (No.: 2019B010155001), and the Program for Guangdong Introducing Innovative and Enterpreneurial Teams (No.: 2017ZT07X183).

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
