# Supplementary Material For "TANGO: Text-driven Photorealistic and Robust 3D Stylization via Lighting Decomposition"

**Yongwei Chen[1,3], Rui Chen[1], JiaBao Lei[1], Yabin Zhang[2], Kui Jia[1,4]***
[1]South China University of Technology [2]The Hong Kong Polytechnic University
[3]DexForce Co. Ltd. [4]Peng Cheng Laboratory
eecyw@mail.scut.edu.cn, kuijia@scut.edu.cn

## A    Additional stylization results

We show more qualitative results of original meshes and downsampled meshes in Figure 1. The original meshes, bird, dragon, car and horse, have 75126, 73568, 64962 and 72494 faces respectively, which are all downsampled to 5000 faces. Note that the similar performance on original and downsampled meshes of our approach verifies our strong robustness. Furthermore, TANGO also performs well in simulating ice effects as shown in the experiment "A bird made of ice", generating surprising transparent effects on both original and downsampled meshes.

## B    User study for low-quality meshes

Besides calculating mean opinion scores on original meshes stylization, we further conduct a user study experiment on downsampled meshes style transfer results. Specifically, we randomly chose 56 users to evaluate 6 downsampled meshes and style text prompts combinations. Each of them was asked three questions: (Q1) "How natural is the output result?" (Q2) "How well does the output match the original content?" (Q3) "How well does the output match the target style?" We report the mean opinion scores with standard deviations for both Text2Mesh and ours in Table 1. TANGO outperforms the Text2Mesh baseline across all questions, with an advantage of 1.28, 1.07 and 1.48 for Q1-Q3, respectively.

Table 1: Mean opinion scores (1-5) for Q1-Q3 (cf. Sec. B), for TANGO and baseline.

|  | (Q1): Overall | (Q2): Content | (Q3): Style |
|---|---|---|---|
| Text2Mesh | $2.91(\pm 0.71)$ | $2.82(\pm 0.80)$ | $2.82(\pm 0.68)$ |
| Ours | $\mathbf{4.19}(\pm 0.43)$ | $\mathbf{3.89}(\pm 0.67)$ | $\mathbf{4.30}(\pm 0.49)$ |

## C    More Ablation results

Besides the ablation study in our main paper, we ablate the backbone network used in CLIP model. As illustrated in Figure 7, the results with transformer-based CLIP present more natural appearance than these based on ResNet.

---

*Corresponding Author

36th Conference on Neural Information Processing Systems (NeurIPS 2022).

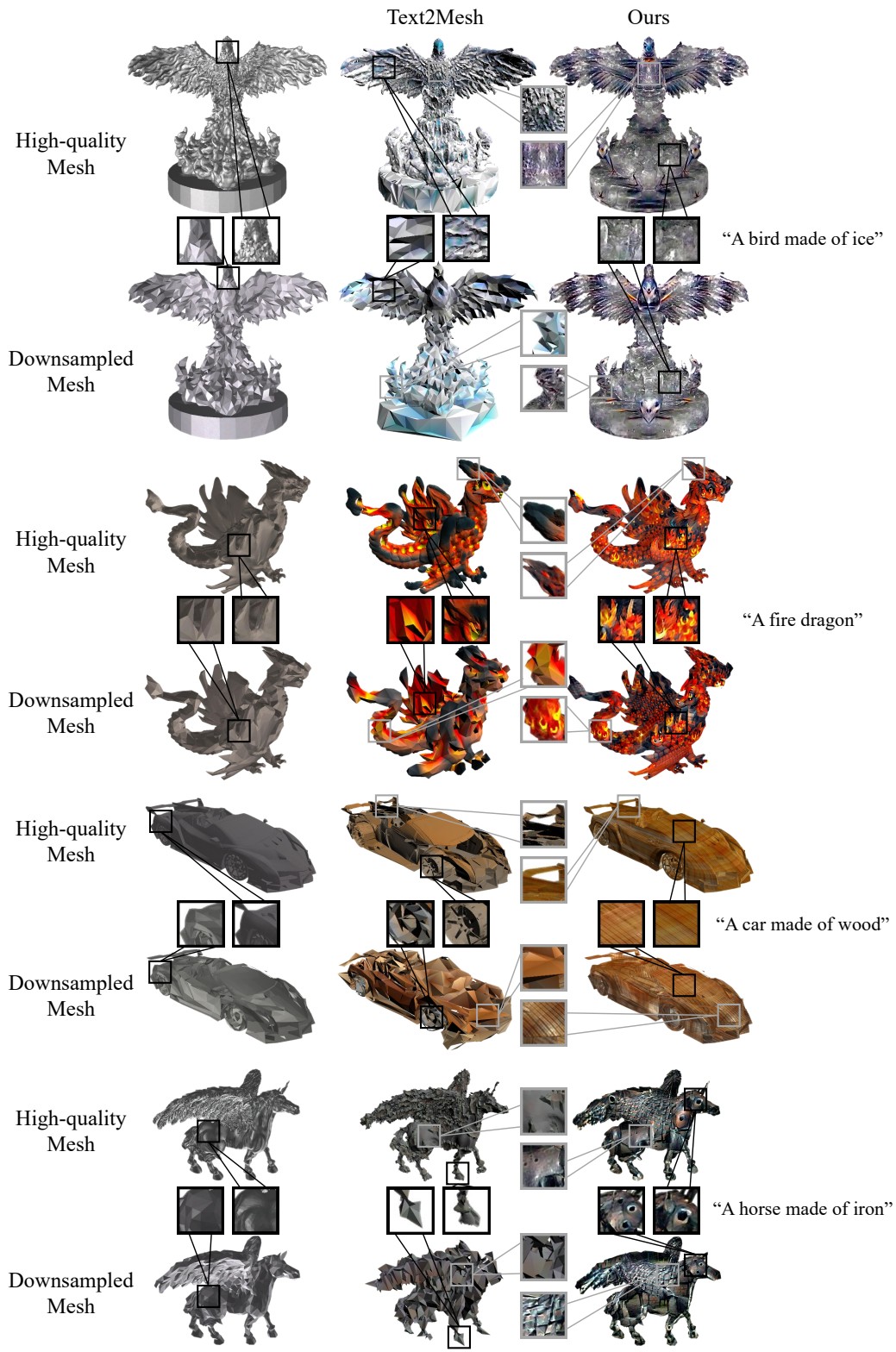

Figure 1: A robustness comparison of TANGO and Text2Mesh on high-quality meshes (cf. the odd rows) and downsampled meshes (cf. the even rows). The results of Text2Mesh degenerates significantly as the mesh quality downgrades, while TANGO presents consistent results on meshes of various qualities.

"A 3D rendering of the Batman in unreal engine"

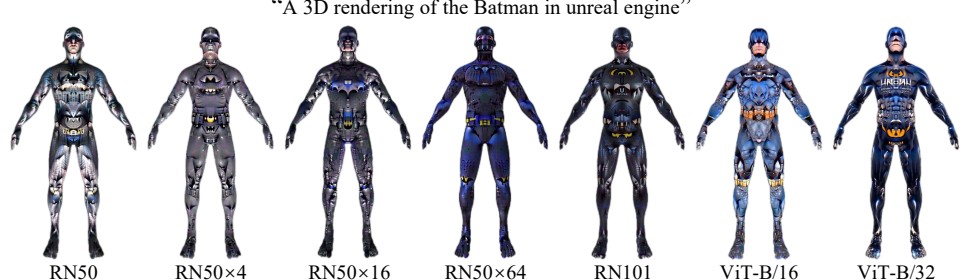

| RN50 | RN50×4 | RN50×16 | RN50×64 | RN101 | ViT-B/16 | ViT-B/32 |

Figure 2: Qualitative results of varying backbone network in CLIP. RN stands for ResNet[2] and ViT represents Vision Transformer [1]. The results with transformer-based CLIP present more natural appearance than these based on ResNet.

# D  An in-depth comparison between TANGO and the vertex displacement framework

Although the vertex displacement framework presents a larger geometry capacity in theory, the rendering capacity gap between frameworks of vertex displacement and our adopted normal displacement could be largely reduced with our introduced learnable SVBRDF and normal. Generally speaking, when we render a pixel $p$, we project a ray from the camera center through the pixel. The intersection point between the ray and geometry surface is denoted as $x_p$ and its corresponding surface normal is denoted as $n_p$. Then, the pixel color is calculated by inputting $x_p$, $n_p$, and the view direction $\nu_p$ to the following rendering equation:

$$L_p(\nu_p, x_p, n_p) = \int_{\Omega} L_i(\omega_i) f_r(\nu_p, \omega_i, x_p)(\omega_i \cdot n_p)\mathrm{d}\omega_i,$$

where $L_i(\omega_i)$ is the incident light intensity from direction $\omega_i$ and $f_r(\nu_p, \omega_i, x_p)$ represents spatially varying BRDF. According to the above rendering equation, the vertex displacement adjusts the pixel color by changing $x_p$ and $n_p$, which will be analyzed individually in the following. Specifically, changing $x_p$ only influences the SVBRDF term $f_r(\nu_p, \omega_i, x_p)$, which is modeled as an MLP network in TANGO. Considering the universal approximation property of neural networks, the MLP-based SVBRDF term in TANGO is expected to cover (at least approximately) the effects of $x_p$ change in the vertex displacement framework. Meanwhile, we refine the term $n_p$ with a normal prediction network, which similarly covers (at least approximately) the $n_p$ change in the vertex displacement framework. Taking it a step further, the possible difference between the vertex displacement framework and TANGO appears in the contour area. In this area, some rays that originally hit the surface may not hit it due to the geometry change via vertex displacements; therefore, the pixels that are originally colorful may turn to the background color, and vice versa. Fortunately, the contour area only occupies a small percentage of the whole rendering images, leading to a small rendering capacity gap between TANGO and the vertex displacement framework.

Just as every coin has two sides, we emphasize that several advantages are achieved by keeping the geometry unchanged. Firstly, more robust results are achieved with unchanged geometry. In the vertex displacement framework, it is difficult to control the displacement direction and self-intersection may occur everywhere with improper vertex displacement, especially when the displacement is significant.

Secondly, TANGO is more robust to the number of vertices (i.e., the mesh quality). As detailed in Sec. 4.2 in our paper, the vertex displacement method requires a large number of vertices and its performance degrades significantly as the vertex number reduces. On the contrary, TANGO is quite robust when mesh quality degenerates and works well with such low-poly meshes, presenting a wide application in industrial 3D assets creation.

Last but not least, keeping the geometry unchanged is more preferred in current game engines. When users want to change the style of an object, it is time-consuming to re-import another geometry and then run the physics simulation again. In contrast, it is quite convenient to replace the material and

normal map for the target rendering style, which is a widely used technique in the AAA games industry.

In conclusion, compared to the vertex displacement framework, TANGO has a slightly smaller capacity but presents more robustness and time efficiency. Considering the advantages and disadvantages of geometry preservation, whether should we preserve the geometry in 3D stylization is still an open problem. TANGO and the recent Text2Mesh conduct the initial attempts to this problem in different directions, which may inspire more in-depth following investigations.