# OpenReview forum: "TANGO: Text-driven Photorealistic and Robust 3D Stylization via Lighting Decomposition"
_NeurIPS.cc/2022/Conference — NeurIPS 2022 Accept_

### Official Review · Reviewer_ojBL · 2022-07-08

**Rating:** 6
**Confidence:** 4
**Soundness:** 4 excellent
**Presentation:** 3 good
**Contribution:** 2 fair

**Summary:**

The paper describes a method for generating a reflectance map (and lighting environment) for a given 3D shape, based on a text prompt, using a CLIP-guided loss.

**Questions:**

Are there any constraints on the normal fitting, e.g., do the normals need to be consistent with the geometry or can they be totally unrelated? Presumably there is a limit to how far normals can vary from the normals of the input geometry (or the nearby vertex normals).



**Limitations:**

One limitation is that stylization may involve creating new geometry, not just texture. "text2mesh" does create new geometry, which has visible pros and cons in the results.

As discussed above, it's unclear whether good materials+lighting have been recovered, or just enough to render views with static lighting.

**Strengths And Weaknesses:**

The problem being tackled is novel, and several of the results are plausibly-good. The impact of the work is incremental, but the general direction of text-guided geometry/texture generation is a significant one.

One big gap in the paper for me is a clear explanation for the motivation for using the SVBRDF/env. map rather than just fitting a texture map. It's clear that the results look better than just fitting a texture map (as shown in the ablations), so this aspect is clear. But, by fitting materials+lighting, the paper gives the possible impression that one gets a good materials+lighting in general, not just for a single view. Is this true? It hasn't been demonstrated or tested. Specifically: can the model be relighted and the viewpoint moved with sensible results? For example, do shiny materials have highlights that move in sensible ways, and diffuse materials do not? etc.

I think the description of the method is unnecessarily mathematical and opaque, with many different terms. However, I don't have any suggestions for improving it and it might be that this is necessary level of detail. Eq 1 has a lot of terms but, in a way, it all boils down to one integral with a bunch of parameters.

Note that "text2mesh" was published at CVPR 2022 after the NeurIPS deadline, so I don't think it "counts" as prior work. Nonetheless, it's great that it's included as a baseline, which is very informative.

---

> ### Author Response · Authors · 2022-08-01
> **Reply to Reviewer ojBL**
>
> We thank reviewer ojBL for the valuable comments, and the point-to-point response to every individual comment is itemized as below:
>
> $Q1.$ **More visualization with moving viewpoint and relighting.**
>
> We appreciate your suggestion. We investigate the moving viewpoint on a shiny object of "A shoe made of gold" and a diffuse object of "A candle made of wicker",
> whose gif results are illustrated in the ANONYMOUS link -- https://anonymous.4open.science/r/NeurIPS_1808/.
> Our method learns good materials and lightings in general and performs well on different views.
> Additionally, as expected, the shiny materials (e.g., gold) have highlights that move in sensible ways, while the diffuse materials (e.g., wicker) do not.
> What's more, our method also performs well on tasks of relighting and material editing, which could be visualized in the $Experiments.pdf$ with the ANONYMOUS link.
> Note that the PDF cannot be illustrated normally on the website, so please download it for better visualization.
>
> $Q2.$ **Complicated equation**
>
> Thanks for your advice. We admit that Eq. 1 is complicated, but we think it is necessary since it provides an overall of our adopted rendering pipeline.
> To help readers understand the equation, we break down the equation into environment map ${L}_i$, BRDF function ${f}_r$ and normal map  $\hat{{n}}_p$, which are explained one by one in line 168 to 191.
> To ease understanding, we may consider replacing the term ${\Pi}({n}_p,{x}_p;{\gamma})$ in Eq. 1 to $\hat{{n}}_p$ in the future revision.
>
> $Q3.$ **Constraints on the normal fitting.**
>
> We indeed set constraints on both normal generation and normal displacements.
> As described in line 174 to 175, we clamp the output normals in $\hat{{n}}_p \in  (1, \hat{\theta}_p, \hat{\varphi}_p) | \hat{\theta}_p \in (0,2\pi), \hat{\varphi}_p \in (0,\pi) $.
> As for the normal displacements $\triangle {n}_p$, we clamp it in $\triangle {n}_p \in (0, \triangle{\theta}_p, \triangle{\varphi}_p) | \triangle{\theta}_p \in (-\frac{\pi}{3},\frac{\pi}{3}), \triangle{\varphi}_p \in (-\frac{\pi}{3},\frac{\pi}{3})$.
>
>
> $Q4.$ **An in-depth comparison between our method and the vertex displacement framework.**
>
> Please refer to the official comment to all reviewers.

---

### Official Review · Reviewer_8TXp · 2022-07-10

**Rating:** 6
**Confidence:** 4
**Soundness:** 3 good
**Presentation:** 3 good
**Contribution:** 3 good

**Summary:**

This paper focuses on the task of 3D mesh stylization according to the text prompt. The key idea is to jointly learn three disentangled components, ie, the spatially varying bidirectional reflectance distribution function, the local geometric variation, and the lighting condition, under the supervision of CLIP Loss with a spherical Gaussians-based differentiable renderer. Experiments show that the proposed method achieves better visualization quality and 3D geometry consistency on stylized meshes compared to the state-of-the-art. Besides, this method demonstrates robustness when stylizing low-quality meshes.

**Questions:**

1. I wonder why the authors adopt “randomly cropped images” rather than “multi-view images” that text2mesh adopts in the phase of data augmentation. Is it because that normal map helps the model focus on the geometry and cropped images guide the model to focus on the detail? If text2mesh adopts cropped images in data augmentation, will it produce more details? An ablation study should be conducted here.
2. What is the purpose of adding a random background to the rendered image in data augmentation? Is it necessary? I do not see the ablation study about the background in experiments.
3. Is it easy to explicitly control the lighting intensity and direction? How should we determine the light distribution?

**Limitations:**

CLIP model may produce semantic ambiguity. It seems that the target text has to contain the category of input mesh for content preservation, which may limit the application of this method in real life.

**Strengths And Weaknesses:**

Strengths:
1. This is an interesting idea that may inspire future research in related fields. It employs a professional shading model to help learn more photorealistic appearance and geometry, which is intuitively reasonable and practically effective.
2. The paper is well-organized and easy to follow.
3. The experiments look convincing since it supports the major contributions claimed in the introduction that the realism of rendered images can be improved by the explicit modeling of light and materials. Moreover, the ablation study also demonstrates that each disentangled component plays an important role in the improvement of the rendering quality.
4. The authors also provide some reasonable limitations of the current method, which may help researchers to build new frameworks based on current results.

Weakness:
1. This paper proposes to learn materials, normals, and light separately. How can **a good disentanglement** of light and physical reflectance be guaranteed? In the whole pipeline, only a CLIP loss is used. It seems like the network here has no motivation to learn a good disentanglement. In Fig4, the relighting results are so weird since the materials are also modified here. How about changing the light only?
2. The quality of 3D stylization results may not be impressive enough in Fig.2. It is not easy to evaluate the performance of the proposed method from the target text prompt. Such as “A shoe made of brick”, I know the proposed method produces more details, but it seems that the result of test2mesh is more natural. A similar situation occurs in "A vase made of colorful crochet". It is hard to judge between this method and Text2Mesh. Of course, this method can preserve better 3D geometry. However, is geometry preservation necessary in 3D stylization? By the way, there are a lot of clear artifacts in the case of "A candle wearing jeans". I think more convincing visualization results can be provided.
3. The quantitative result in Table 1 is unclear. As this work splits the style into geometry and appearance and claims the improvement in both, user studies should be conducted separately in terms of both aspects.

---

> ### Author Response · Authors · 2022-08-01
> **Reply to Reviewer 8TXp**
>
> We thank reviewer 8TXp for the valuable comments, and the point-to-point response to every individual comment is itemized as below:
>
> $Q1.$ **The key point for a good disentanglement.**
>
> We think the key point is the shading model and carefully designed elements in Eq. 1 in our paper, which is a strong physical prior for learning parameters.
> By predicting normal offsets and SVBRDF individually, optimizing the light SG parameters, and inputting these elements into the physical rendering equation,
> the individual module predicts physics-aware results, as illustrated with the disentangled components in the $Experiments.pdf$ at https://anonymous.4open.science/r/NeurIPS_1808/.
> Note that the PDF cannot be illustrated normally on the website, so please download it for better visualization.
> What's more, it is also important to clamp the networks' output in a reasonable range,
> e.g., clamping the predicted normal $\hat{{n}}_p = (1, \hat{\theta}_p, \hat{\varphi}_p)$ in $\hat{\theta}_p \in (0,2\pi), \hat{\varphi}_p \in (0,\pi) $.
>
>
> $Q2.$ **The clarification and visualization on relighting and material editing.**
>
> In Figure 4, we only change the lighting and keep the material fixed. We are will clarify the misleading and we will make it more clear in the revision.
> Furthermore, we provide more visualizations about relighting and material editing in the $Experiments.pdf$, which could be accessed through an ANONYMOUS link -- https://anonymous.4open.science/r/NeurIPS_1808/.
> Note that the PDF cannot be illustrated normally on the website, so please download it for better visualization.
> These visualizations demonstrate the ability of our method on the rendering component disentanglement, which will be added in our future revision.
>
> $Q3.$ **Illustration with more convincing results.**
>
> Thanks for your advice. We appreciate that you find our method producing more details and preserving better 3D geometry. We will provide more convincing results in the revision and in our future webpage.
>
> $Q4.$ **An in-depth analysis of the geometry preservation for 3D stylization.**
>
> Please refer to the official comment to all reviewers: "An in-depth comparison between our method and the vertex displacement framework".
>
> $Q5.$ **New user study in geometry and appearance.**
>
> Thanks for your advice. We conduct a new user study by investigating geometry and appearance individually.
> We randomly chose 127 users to evaluate 13 source meshes and style text prompt combinations on both high and low-quality meshes. Each of them was asked two questions:
> (Q1) "How well does the output geometry match the text prompt?" (Q2) "How well does the output appearance effects match the text prompt?".
> Our method outperforms the Text2Mesh baseline on both questions, with an advantage of 0.71 and 0.92 for Q1 and Q2, respectively. We will add this new user study in the revision.
>
> $Q6.$ **The clarification on the data augmentation.**
>
> Indeed, our data augmentation strategy is the same as that adopted in Text2Mesh.
> Specifically, we use multiview images and randomly cropped images simultaneously to ensure the global semantic consistency and preserve local details,
> which is identical to the description in Sec. 3.3 of Text2Mesh paper and their official code repository.
> We will clarify this part more clearly in the future revision.
>
> $Q7.$ **The clarification on the random background.**
>
> Indeed, our method is insensitive to the background.
> The reason is that we only render those pixels whose rays hit the object's surface, and then the intersection point and its normal will feed into the network.
> For the background pixels whose rays do not hit the surface, though we assign a background value to them, they are not input to the network and their gradients are not backpropagated to update the network parameters.
> We test black, white, and random Gaussian backgrounds, and observe similar stylized performance. We will add these analyses in the revision.
>
>
> $Q8.$ **The Controlling of light intensity, direction, and light distribution.**
>
> Thanks to our spherical Gaussian representation, we can easily control light.
> Users can change the light intensity and light direction by modifying ${a}_k$ and ${\mu_k}$ in multiple spherical Gaussians of the environment map, respectively.
> Meanwhile, users can easily modify the light distribution by modifying the spherical Gaussians or adopting a new environment map from the Internet, which is shown in our additional relighting experiment in the $Experiments.pdf$.
>
>
> $Q9.$ **Semantic ambiguity of CLIP model.**
>
> We agree with your opinion. The major reason for this phenomenon is the capability of the CLIP model.
> Fortunately, our method has the potential to be guided by other Vision-language Pre-training models.
> The semantic ambiguity of the CLIP model may be alleviated with more powerful Vision-language Pre-training models.

---

> > ### Comment · Reviewer_8TXp · 2022-08-09
> > **Response to the authors**
> >
> > Thanks for the comments and additional experiments. The authors' response has resolved most of my concerns, especially the explanation of the disentanglement of light and reflectance. On the other hand, I agree with the comments from Review kpuS that the limitations of this method and Text2mesh should be discussed carefully in the revision. Besides, More pure geometry results should be presented since it's claimed that this method could stylize both 3D shape and appearance. Meanwhile, the authors should also release the meshes and the text prompts for the user study conducted on geometry and appearance individually. Finally, I encourage the authors to provide more convincing results as promised. After thorough consideration, I decide to keep my original rating.

---

### Official Review · Reviewer_kpuS · 2022-07-11

**Rating:** 4
**Confidence:** 4
**Soundness:** 3 good
**Presentation:** 3 good
**Contribution:** 2 fair

**Summary:**

This paper presents a 3D mesh stylization method that performs both appearance and geometric styles by optimizing with a differentiable rendering pipeline. Compared with existing methods that simply optimize per-vertex color and displacement, it adopts a more physics-aware rendering model that explicitly models geometry (normal map), BRDF (diffuse, roughness, specular), and lighting (SG), which is claimed to generate more realistic results (better shading appearance and details) and allows for editing such as relighting. Results are compared with Text2Mesh.

**Questions:**

I have put my major concerns in the weaknesses section. Overall, I think it is an interesting work that combines neural rendering with mesh stylization that could produce meaningful assets that are more compatible with current rendering pipelines and allow for more wide editing applications. However, the presented results, unfortunately, do not fully the potential of the proposed method -- there are no results showing the quality of individual components and the possibility to edit one of them for editable stylization, without these I am not convinced that this method has achieved significant improvement over Text2Mesh except some better shading details. Also considering that this method is not able to produce real geometric details, and the visual results on single stylization images are not clearly better than Text2Mesh, I cannot say this method advances the state-of-the-art. Finally, given the limited technical contributions, I am leaning a bit negative unless more convincing evidence could be presented.

**Limitations:**

There are discussions on limitations.

**Strengths And Weaknesses:**

++ Indeed very promising idea to do physics-aware modeling of mesh stylization.

++ Interesting visual results are presented, which look good.

++ The proposed rendering formulation is technically sound and practically reasonable.

++ Very good paper-writing, except for some redundancy that could be made more concise.

-- All results are just final renderings, lacking results to show the individual components of normal maps, BRDFs, and lightings, which are critical to show the quality of the system and its advantage over Text2Mesh.

-- No results on manipulating the rendering, such as relighting (the relighting results on the shoe are less interesting for stylization) and material editing, which are not achievable by Text2Mesh.

-- Approximating geometry with normal maps turns the method into a pure appearance stylization method. There are no actual geometry changes but just a way to model geometry variation for shading. But Text2Mesh does achieve joint geometry and appearance stylization, which seems better than this method although the geometric details have limitations.

-- Limited contributions, the rendering pipeline, and CLIP-guided stylization are mostly based on existing works.

---

> ### Author Response · Authors · 2022-08-01
> **Reply to Reviewer kpuS**
>
> We thank reviewer kpuS for the valuable comments, and the point-to-point response to every individual comment is itemized as below:
>
> $Q1.$ **Analyses on individual components, material editing, and relighting.**
>
> We thank the reviewer for pointing out this issue.
> We provide more analyses in the $Experiments.pdf$ file with this ANONYMOUS link -- https://anonymous.4open.science/r/NeurIPS_1808/.
> Note that the PDF cannot be visualized normally on the website, so please download it for better visualization.
>
> Specifically, in Figure 1 of the $Experiments.pdf$, we visualize the individual components of the normal map, BRDF, and environment map with the example of "A shoe made of brick". These correctly estimated components contribute to the realism of final renderings, and furthermore, make it possible to edit the generated stylization, which is shown as follows.
>
> In Figure 2, we conduct the material editing with the example of "A shoe made of gold.".
> The surface becomes more diffuse as we enlarge the roughness value of the material, while a larger specular value typically results in a more shiny surface, proving the capability of our method on material editing.
> In Figure 3, we conduct the relighting experiments with the example of "A vase made of wood".
> By replacing the estimated environment map with others downloaded from the Internet, the lighting of the rendering results could be successfully manipulated.
> Compared to the Text2Mesh, our method presents more flexibility in manipulating the rendering results, facilitating its application in practice.
> We will add these analyses in the revision.
>
>
> $Q2.$ **An in-depth comparison between our method and Text2Mesh.**
>
> Please refer to the official comment to all reviewers.
>
> $Q3.$ **A clarification on the contribution.**
>
> We agree with the reviewer that the rendering pipeline and the CLIP-guided stylization have been studied in existing works, and we do not claim these as our contributions.
> Actually, our major contribution is a text-driven 3D stylization architecture, which can transfer the appearance style of a given 3D shape according to a text prompt in a photorealistic manner.
> Compared to the recent Text2Mesh which utilizes vertex displacement, we investigate a different technical approach and achieve higher photorealistic quality and more robustness to the mesh quality.
> A more detailed comparison between our method and the Text2Mesh could be found in the official comment to all reviewers.
>
> From another perspective of rendering components disentanglement, we make the first attempt to predict and disentangle the rendering components (e.g., normal map, SVBRDF, and environment map) under quite weak supervision, e.g., a short text prompt,
> which is typically investigated under the strong pixel-by-pixel image supervision in previous works, e.g., NeRD[4], PhySG[53], and nvdiffrec[36].

---

> > ### Comment · Reviewer_kpuS · 2022-08-08
> > **Thanks for the response**
> >
> > Thanks Authors for the response. I appreciate the additional results, which indeed helps demonstrate the capability for meaningful rendering edits besides only showing the final results. While I am still not fully convinced if the proposed mesh stylization method is better than Text2Mesh, I am not against acceptance, but would suggest the authors to make it more clear the difference to Text2Mesh and highlight the limitations of both methods. In addition, I recommend to rephrase the title and the major claim that this is a method for "3D Stylization For Arbitrary Meshes", which may imply that this stylize both 3D shape and appearance. And please also try to include more rendering-aware manipulation results, like those shown in the new supp, to the final version.

---

> > > ### Author Response · Authors · 2022-08-09
> > > **Thanks for your reply**
> > >
> > > We thank you again for the constructive comments and recognition of our method. In the final version, we will follow the suggestions by clearly summarizing the difference between Text2Mesh and our method, and presenting the limitations of both methods. We would also revise the title to "Text-driven Photorealistic and Robust 3D Stylization via Lighting Decomposition", if all reviewers agree with the title as well; we will correspondingly revise the major claim. Finally, we will add more rendering-aware manipulation results like decomposition, relighting, and material editing in our final version.

---

### Official Review · Reviewer_qBos · 2022-07-11

**Rating:** 5
**Confidence:** 2
**Soundness:** 3 good
**Presentation:** 3 good
**Contribution:** 3 good

**Summary:**

This paper targets at stylizing a mesh with a text prompt. It represents appearance of a mesh as spatially varying BRDF, normal and lighting, and matches the CLIP feature of rendered image and given text prompt by learning the parameters of these components. For evaluation, the paper compares with the state-of-the-art text prompt stylization method Text2Mesh and achieves better performance.

**Questions:**

**Method.**
I am actually surprised this model can predict correct lighting parameters given only CLIP latent space as supervision. Have the authors observed any ambiguity in disentangling lighting and reflectance? If not, what is the key to make the model learn correct disentanglement?

**Implementation details.**
- How to find reasonable anchor view in line 145?
- How to find the first intersection point and intersection face in line 150?

**Novelty.**
Although not using text prompt as supervision, a similar lighting model has been explored in the NeRD [4] paper, while the text prompt stylization has been used by Text2Mesh. Thus although the idea is interesting, the novelty and contribution of the proposed method is limited, in my opinion.

**Limitations:**

The authors have discussed the limitations of the chosen lighting model. As stated in the weakness, I encourage the authors to include
 the limitation of not modeling geometric displacement.

**Strengths And Weaknesses:**

**Strengths.**
- This paper is well written and easy to follow.
- The idea of representing the appearance of a given mesh as SVBRDF, normal and lighting is interesting, although its has certain limitations.

**Weaknesses.**
- Modeling appearance as a lighting model certainly performs better in complex realistic scenes. However, it also limits the capacity of the model without geometric displacements modeling. For instance, in Fig.3 in the Text2Mesh paper, the surface of the given mesh can be changed to match the text prompt, while the proposed method cannot do so. This limitation is not discussed in the paper.

---

> ### Author Response · Authors · 2022-08-01
> **Reply to Reviewer qBos**
>
> We thank reviewer qBos for the valuable comments, and the point-to-point response to every individual comment is itemized as below:
>
> $Q1.$  **The advantages and disadvantages of geometry preservation in 3D stylization.**
>
> Please refer to the official comment to all reviewers.
>
> $Q2.$ **The key to learning correct disentanglement.**
>
> Actually, we do not observe any ambiguity in disentangling lighting and reflectance.
> We think the key point is the shading model and carefully designed elements in Eq. 1 in our paper, which set a strong prior for learning parameters.
> By predicting normal offsets and SVBRDF with two individual networks, optimizing the light SG parameters, and inputting these elements into the rendering equation, each module is constrained to predict physics-aware results.
> Moreover, it is also important to constrain the networks' output in a reasonable range,
> e.g., clamping $\hat{\theta}_p \in (0,2\pi)$ and $\hat{\varphi}_p \in (0,\pi)$ for the normal prediction $\hat{{n}}_p = (1, \hat{\theta}_p, \hat{\varphi}_p)$.
>
> $Q3.$ **The definition of a reasonable anchor view.**
>
> Except for the person object, where a front view is adopted as the anchor, we randomly sample the anchor view in $(r, \theta, \phi)$ with $\theta \in (-\pi,\pi)$ and $\phi \in (0,2\pi)$.
>
> $Q4.$ **The implementation of the first intersection point and intersection surface.**
>
> We use the ray casting method implemented in the python Open3d library to find the first intersection point and intersection surface.
>
> $Q5.$ **A comparison between NeRD and our method.**
>
> We have clarified the difference between our method and the vertex displacement framework (e.g., Text2Mesh) in the official comment to all reviewers.
> In the following, we will illustrate the difference between our method and NeRD, which has been briefly discussed in our related work.
>
> First of all, NeRD and our method are proposed for different tasks. NeRD reconstructs 3D assets (e.g., 3D meshes and lighting) from multiple images, which provide strong pixel-by-pixel supervision signals.
> In contrast, we aim to generate photorealistic 3D stylization given arbitrary meshes and a text description.
>
> Taking it a step further, NeRD and our method share similar construction objectives if we view it from a more general perspective of 3D assets construction.
> However, as detailed in the first point, the input of the two methods are quite different.
> Additionally, it takes about 144 GPU hours and 1.5 GPU hours to train a NeRD and extract 3D assets, respectively.
> In contrast, our method only consumes about 0.2 GPU hours to get a new 3D asset, presenting a significant advantage in time efficacy.

---

### Author Response · Authors · 2022-08-01
**For all reviewers and ACs**

$Q1.$  **An in-depth comparison between our method and the vertex displacement framework.**

Although the vertex displacement framework presents a larger geometry capacity in theory,
the rendering capacity gap between frameworks of vertex displacement and our adopted normal displacement could be largely reduced with our introduced learnable SVBRDF and normal.
Generally speaking, when we render a pixel $p$, we project a ray from the camera center through the pixel.
The intersection point between the ray and geometry surface is denoted as $x_p$ and its corresponding surface normal is denoted as $n_p$.
Then, the pixel color is calculated by inputting $x_p$, $n_p$, and the view direction $\nu_p$ to the following rendering equation:

$$
L_p(\nu_p, x_p, n_p) = \int_\Omega L_i({\omega}_i)f_r({\nu}_p,{\omega}_i,{x}_p)({\omega}_i \cdot {n}_p ) \mathrm{d}\omega_i,
$$

where $L_i({\omega}_i)$ is the incident light intensity from direction $\omega_i$ and $f_r({\nu}_p,{\omega}_i,{x}_p)$ represents spatially varying BRDF.
According to the above rendering equation, the vertex displacement adjusts the pixel color by changing $x_p$ and $n_p$, which will be analyzed individually in the following.
Specifically, changing $x_p$ only influences the SVBRDF term $f_r({\nu}_p,{\omega}_i,{x}_p)$, which is modeled as an MLP network in our method.
Considering the universal approximation property of neural networks, the MLP-based SVBRDF term in our method is expected to cover (at least approximate) the effects of $x_p$ change in the vertex displacement framework.
Meanwhile, we refine the term ${n}_p$ with a normal prediction network, which similarly covers (at least approximates) the ${n}_p$ change in the vertex displacement framework.
Taking it a step further, the possible difference between the vertex displacement framework and our method appears in the contour area.
In this area, some rays that originally hit the surface may not hit it due to the geometry change via vertex displacements; therefore, the pixels that are originally colorful may turn to the background color, and vice versa.
Fortunately, the contour area only occupies a small percentage of the whole rendering images, leading to a small rendering capacity gap between our method and the vertex displacement framework.

Just as every coin has two sides, we emphasize that several advantages are achieved by keeping the geometry unchanged.
Firstly, more robust results are achieved with unchanged geometry.
In the vertex displacement framework, it is difficult to control the displacement direction and self-intersection may occur everywhere with improper vertex displacement, especially when the displacement is significant.

Secondly, our method is more robust to the number of vertices (i.e., the mesh quality).
As detailed in Sec. 4.2 in our paper, the vertex displacement method requires a large number of vertices and its performance degrades significantly as the vertex number reduces.
On the contrary, our method is quite robust when mesh quality degenerates and works well with such low-poly meshes, presenting wide applications in industrial 3D assets creation.

Last but not least, keeping the geometry unchanged is more preferred in current game engines.
When users want to change the style of an object, it is time-consuming to re-import another geometry and then run the physics simulation again.
In contrast, it is quite convenient to replace the material and normal map for the target rendering style, which is a widely used technique in the games industry.

In conclusion, compared to the vertex displacement framework, our method has a slightly smaller capacity but presents more robustness and time efficiency.
Considering the advantages and disadvantages of geometry preservation, whether should we preserve the geometry in 3D stylization is still an open problem.
Our method and the recent Text2Mesh conduct the initial attempts to this problem in different directions, which may inspire more in-depth following investigations.

---

### Comment · Area_Chair_eZBJ · 2022-08-09
**reminder for discussion**

Dear reviewers,

Thank you all for providing valuable comments. The authors have provided detailed responses to your comments. Has the response addressed your major concerns?

If you haven't, It would be great if you could reply to the authors’ responses soon as the deadline is approaching (Tues, Aug 9).

Best,

ACs

---

### Meta-Review · Area_Chair_eZBJ · 2022-08-29

**Recommendation:** Accept
**Confidence:** Certain

**Metareview:**


This paper presents a new CLIP-driven stylization method given an input mesh and text description. Compared to previous works Text2Mesh, the paper introduces a more expressive rendering model based on learnable SVBRDF and normal maps. Many reviewers found the paper easy to follow, the idea promising, and the results visually appealing. They also expressed their concerns regarding the similarity to Text2Mesh, the limitations of the normal maps approach (compared to changing geometry explicitly), and the relighting and material editing of the stylized object. The rebuttal has addressed most of the concerns. The AC agreed with most of the reviewers and recommended accepting the paper.

Please revise the papers according to the reviewer’s comments: (1) change the title according to Reviewer kpuS, (2) add relighting/material editing/view synthesis results, and (3) highlight the pros and cons of the proposed method w.r.t. Text2Mesh.



**Award:**

No

---

### Decision · Program_Chairs · 2022-09-14

Accept